# Spatial Transcriptomics Reveals Novel Mechanisms Involved in Perineural Invasion in Pancreatic Ductal Adenocarcinomas

**DOI:** 10.3390/cancers17050852

**Published:** 2025-03-01

**Authors:** Vanessa Lakis, Noni L Chan, Ruth Lyons, Nicola Blackburn, Tam Hong Nguyen, Crystal Chang, Andrew Masel, Nicholas P. West, Glen M. Boyle, Ann-Marie Patch, Anthony J. Gill, Katia Nones

**Affiliations:** 1QIMR Berghofer Medical Research Institute, Brisbane, QLD 4029, Australia; vanessa.lakis@qimrb.edu.au (V.L.); tamhong.nguyen@qimrb.edu.au (T.H.N.); crystal.chang@qimrberghofer.edu.au (C.C.); andrew.masel@qimrb.edu.au (A.M.); glen.boyle@qimrb.edu.au (G.M.B.); devonpatches@gmail.com (A.-M.P.); 2NSW Health Pathology, Department of Anatomical Pathology, Royal North Shore Hospital, Sydney, NSW 2065, Australia; noni.chan@health.nsw.gov.au (N.L.C.); anthony.james.gill@sydney.edu.au (A.J.G.); 3Australian Pancreatic Cancer Genome Initiative (APGI), Kinghorn Cancer Centre, Sydney, NSW 2010, Australia; r.lyons@garvan.org.au (R.L.); nicola.blackburn@garvan.org.au (N.B.); 4Griffith Health, Griffith University, Gold Coast, QLD 4215, Australia; n.west@griffith.edu.au; 5School of Biomedical Sciences, The University of Queensland, Brisbane, QLD 4072, Australia; 6School of Biomedical Sciences, Faculty of Health, Queensland University of Technology, Kelvin Grove, QLD 4000, Australia; 7Faculty of Medicine and Health, University of Sydney, Sydney, NSW 2050, Australia; 8Cancer Diagnosis and Pathology Group, Kolling Institute of Medical Research, Royal North Shore Hospital, Sydney, NSW 2065, Australia; 9Faculty of Health, Medicine and Behavioural Sciences/PA Southside Clinical Unit, The University of Queensland, Brisbane, QLD 4102, Australia

**Keywords:** perineural invasion, spatial transcriptomics, *MGLL*, *SAT1*, nerve, pain

## Abstract

Pancreatic cancer has a low survival rate with limited treatment options. Perineural invasion (PNI), where cancer cells infiltrate the nerves in the pancreas, is associated with a poor prognosis and local recurrence. Pancreatic cancer is one of the most painful cancers, with PNI contributing to the pain experienced by patients. However, the mechanisms involved in the cancer invasion of nerves are not completely understood. Here, we used spatial transcriptomics to study the expression of both cancer and nerve cells in microscopic regions of PNI and compared it to regions without PNI evidence to better understand the mechanisms involved in PNI. We identified novel mechanisms that may impact cancer growth, PNI and pain experienced by patients.

## 1. Introduction

Pancreatic ductal adenocarcinoma (PDAC) is a lethal malignancy, with most patients (~80%) being diagnosed at late-stage inoperable disease and with a 5-year survival rate of less than 10% even amongst operable patients [1]. Pancreatic cancer incidence is rising at a rate of 0.5% to 1.0% per year, and it is predicted to become the second leading cause of cancer-related mortality by 2030 [2]. Currently, available therapies have only modest effects on patients’ outcomes [2]. These facts highlight the urgency to better understand the PDAC microenvironment to identify novel treatments for this deadly disease.

Perineural invasion (PNI) is a common pathological feature of PDAC defined as the infiltration of cancer cells into the epineural, perineural and/or endoneural layers of nerves [3]. In PDAC, PNI is found in 71–98% of cases [4], which is amongst the highest frequency across all malignancies. It has been suggested that the high incidence of PNI in PDAC could itself simply reflect the aggressive properties of the cancer. However, PNI in PDAC has been reported without lymphatic or hematogenous spread, with survival beyond three years after surgery restricted to patients without evidence of PNI [5]. Patients with PNI had a four times higher risk of death than those without PNI [6], suggesting that the interactions between cancer and nerve cells play a role in tumorigenesis. Interactions between nerve and cancer cells involve two main mechanisms: PNI and tumor innervation (where axons extend into the TME) [7]. Tumor innervation [8] can be induced by cancer cells through the secretion of neurotrophins such as nerve growth factor (NGF), brain-derived neurotrophic factor (BDNF) and glial cell-derived neurotrophic factor (GDNF), amongst others [3]. Axonogenesis signaling pathways, such as axon pathfinding, have been implicated in PDAC pathogenesis [9]. We showed that ROBO-SLIT Signaling, a negative regulator of the axon guidance pathway, is frequently mutated [10] and methylated in PDAC [11]. Furthermore, there is in vitro evidence that neurotransmitters can enhance cancer growth and migration [5]. The inhibition of the nerve growth factor (NGF)–neurotrophic tyrosine kinase receptor (NTRK1) pathway has been shown to inhibit cancer proliferation, reduce innervation and increase the overall survival of mice when treated together with gemcitabine [12]. These and other studies suggest that crosstalk between nerves and cancer cells drives the high incidence of PNI in PDAC.

PNI in PDAC has been associated with a higher risk of recurrence, poor prognosis [13] and pain severity [4]. Studies have identified potential nerve microenvironment factors involved in PNI; however, the precise mechanisms that lead to nerve invasion and its contribution to cancer pain have yet to be elucidated [14]. A recent report suggested that PNI may contribute to an immune-suppressed microenvironment in PDAC, potentially reducing benefits from immunotherapies [15].

Most of what we know about PNI biology has been obtained using animal and cell models [7], which are key to the study of PNI in PDAC and cancer pain [5]. However, cancer–nerve interactions can occur directly via cancer–nerve cell crosstalk or indirectly via other cells within the TME whereby in vivo/in vitro models lack the complexity to fully understand these interactions [7,16]. Spatial transcriptomics allows for the quantitative measurement of the transcriptome, in particular, the morphological features of the TME, which has the potential to transform our understanding of PNI, reveal the role of the nervous system in cancer and identify novel therapeutic strategies for cancer treatment and pain management in PDAC.

Spatial transcriptomics has been used to study PNI in oral squamous cell carcinomas [17] and a PDAC mouse model [18]. These studies evaluated nerves with and without PNI, indicating that nerves within the TME had expression changes in the genes involved in nerve injury response pathways when compared to nerves outside the TME. Spatial transcriptomics has also been used in pancreatic cancer to study the impact of neoadjuvant therapy on cancer cells [19] and the biology of pre-cancer lesions [20] and more recently to evaluate cancer transcriptional heterogeneity [21,22]. One study [21] used laser micro-dissected cancer cells from different morphological features, gland-forming and non-gland-forming, to identify four transcriptional programs represented across all patients. The authors investigated the expression of these transcriptional programs in cancer cells within the PNI microenvironment and showed enrichment in two transcriptional programs. The authors suggested that thick collagen fibers in the epineurium drive transcriptional changes in cancer cells by contact. While this is a valid assumption, the authors also suggested that transcriptional programs may be driven by microenvironmental signals acting on cancer cells, and we hypothesize that nerves may provide signals that could influence cancer cells transcriptional programs.

Here, we investigated the gene expression of both nerve and cancer compartments in PNI and non-PNI foci in PDAC patients’ samples using spatial transcriptomics. Our aims were to identify novel mechanisms involved in PNI and to generate evidence for further studies to explore new biological processes related to PNI. We identified aspects involved in the complexity of the PNI microenvironment that could impact interactions between nerve and cancer cells and possibly transform treatment and benefit PDAC patients in the future.

## 2. Materials and Methods

Samples

Formalin-fixed paraffin-embedded (FFPE) blocks from PDAC cases were selected from the Australian Pancreatic Cancer Genome Initiative biobank (patients recruited between 2010 and 2013). This study received ethics approval from QIMR Berghofer Research Ethics (P3462).

PDAC cases with a PNI diagnosis had hematoxylin and eosin (H&E)-stained slides reviewed by a pathologist (NLC or AJG) to identify cases with suitable morphological features for spatial transcriptomic analysis. Blocks containing PNI where cancer invaded the nerves inside or outside the main tumor mass and non-PNI regions were selected for this study. Most cases selected had both uninvolved and involved nerves in the same sections. The aim was to collect features within each slide to use patient/slide as a co-variant in the model to account for the tumor heterogeneity between patients.

Digital spatial profiling

For cases identified as suitable for spatial analysis, two consecutive tissue sections (5 μm) were freshly cut, the first for H&E staining to confirm morphology and the second mounted on Superfrost slides (Thermo Fisher, Seventeen Mile Rocks, QLD, AU) for spatial profiling. Spatial profiling was undertaken on the NanoString GeoMx™ Digital Spatial Profiling (DSP) platform at the Central Facility for Genomics at Griffith University (Gold Coast, Australia). Slides were processed following the GeoMx DSP slide prep user manual (MAN10087-04). Briefly, tissue sections were stained with a nuclear stain (Syto13), an epithelial cell marker—pan-cytokeratin (PanCK)—and a pan-neural marker protein gene product 9.5 (PGP9.5). The images of adjacent H&E- and morphology marker-stained slides (GeoMx-DSP) were reviewed together by a pathologist (NLC or AJG) to ensure that accurate morphological features were selected to collect expression tags in regions of interest (ROIs) or areas of illumination (AOIs), where ROIs were further segmented based on the prevalence of the morphology markers. Therefore, when tags were collected from the entire (full) ROIs, gene expression counts may represent expression from all cells in a biological compartment (region). When tags were collected from AOIs, gene expression would be enriched from cells associated with the morphology markers.

Within the selected ROIs or AOIs, the gene expression of over 18,000 targets was assessed using the NanoString (Seattle, WA, USA)—GeoMx Human Whole Transcriptome Atlas (GeoMx Human WTA). We profiled 14 slides from 13 patients and collected 151 ROIs/AOIs, including 76 AOIs segmented by PanCK positivity, 42 AOIs segmented by PGP9.5 positivity and 33 full ROIs in nerves (Appendix A). Samples were sequenced on a NextSeq 2000 and data processed with DRAGEN_GeoMx_3.8.4-96 and imported into DSP-GeoMx to obtain raw tag counts.

QC and preliminary analysis indicated the need to remove ROIs/AOIs that were either too small or too large to reduce the variability in size, which resulted in unmatched PNI/non-PNI regions within slide/patient for the analysis of the nerve compartment (Appendix A). Smaller ROIs resulted in smaller library sizes in PNI compared with non-PNI regions (Appendix A). A total of 40 AOIs segmented by PanCK (cancer compartment) and 17 non-segmented full ROIs (collected in PGP9.5-positive areas) for the nerve compartment (Appendix A) were normalized independently to identify the differentially expressed genes of the cancer and nerve compartments in PNI vs. non-PNI foci.

Data analysis

GeoMx-DSP raw counts (Appendix A) were imported into the standR package (version 1.4.2) [23] using R (version 4.3.1) and transformed into log-counts-per-million (logCPM). For cancer AOIs (segmented by PanCK positivity), logCPM was corrected for batch using the standR geomxBatchCorrection function using the RUV4 method with 500 negative control genes identified through the findNCGs function. Weight matrices from RUV4 batch correction and patient (slide) were used as co-variates when performing differential expression analysis using limma-voom. For the nerve compartment (ROIs), patient/slide was used for batch effect correction, using 400 negative control genes, and the weight matrices were used as co-variates in differential gene expression analysis. Genes were considered differentially expressed if the fold change (FC) was >1.4 and the adjusted *p*-value (Benjamini–Hochberg) was <0.20.

Pathway analysis

Differentially expressed genes were imported into QIAGEN Ingenuity Pathway Analysis (QIAGEN Inc. on 18/07/2024) to identify significantly enriched pathways (Fisher Exact test, *p* ≤ 0.05).

Immunohistochemical analysis

Immunohistochemistry (IHC) was performed using Tyramide signal amplification (TSA) (Akoya Biosciences, Marlborough, MA, USA). FFPE sections (4 µm) were deparaffinized and rehydrated in a Leica XL autostainer. Endogenous peroxidase activity was quenched (2% hydrogen peroxide) for 10 min.

MGLL/PanCK: Antigen retrieval for the MGLL antibody was performed with Dako (pH 6.0) in a Biocare Medical Decloaker chamber (Biocare Medical, Pacheco, CA, USA) at 100 degrees for 20 min. Slides were blocked with a Biocare Medical background sniper for 15 min to reduce non-specific binding. The primary antibody—MGLL (ab307162)—was applied at room temperature. The secondary antibody from Biocare Medical MACH—1 universal HRP—was incubated at room temperature. Slides were rinsed and immersed in antigen retrieval solution from Dako (pH 9.0) buffer in a Biocare Medical Dcloaker chamber at 100 degrees for 20 min. Slides were blocked with a Biocare Medical background sniper for 10 min. The primary antibody pan-cytokeratin (PanCK (AE1/AE3) Dako M3515) was applied at room temperature. The secondary antibody anti-mouse IgG (Goat) HRP (Perkin Elmer) was incubated at room temperature. TSA Opal 650 (PanCK) and Opal 570 (MGLL) were quenched and developed.

Nestin/S100: Antigen retrieval was performed in citrate buffer (pH 6.0) using a Biocare Medical Decloaker chamberat 90 °C for 15 min. Non-specific binding sites were blocked with a Biocare Medical Background Sniper for 10 min. The primary antibody, S100 (Novocastra NCL-L-S100p), was applied at room temperature, followed by incubation with the secondary antibody, anti-rabbit IgG (goat) conjugated with horseradish peroxidase (HRP—PerkinElmer). After rinsing, a second antigen retrieval was conducted in citrate buffer (pH 6.0) at 125 °C for 5 min to prepare for the primary antibody Nestin (Santa Cruz Biotechnology 10C2, sc23927), which was applied at room temperature. Secondary detection was carried out using anti-mouse IgG (goat) HRP (PerkinElmer, Glen Waverley, VIC, AU). TSA Opal dyes were used for signal development: Opal 570 (S100) and Opal 520 (Nestin). Nuclei for all slides were counterstained with DAPI for 5 min and cover-slipped using Dako Fluorescent Mounting Medium (S302380-2)(Dako, Carpinteria, CA, AUSA).

Stained slides were imaged using an Aperio FL slide scanner (Leica Biosystems, Mount Waverley, VIC, AU) at 20× magnification. Optimal exposure settings were determined, and all samples were scanned using the same exposure settings. The immunoreactivity of MGLL, PanCK, Nestin and S100 was evaluated using QuPath (v0.5.1). PNI and non-PN regions were identified in H&E sections and manually annotated on the stained images in QuPath. Regions were then subjected to a pixel threshold classifier to detect areas positive for PanCK or S100 staining. MGLL staining intensity was quantified in PanCK-positive regions in the cancer compartments in PNI and non-PNI foci. Nestin staining was quantified in S100-positive regions in nerves with and without PNI evidence. Additionally, single-cell detection was performed based on DAPI-stained nucleus segmentation followed by the application of a thresholding algorithm to classify cells detected for MGLL or Nestin positivity in PNI and non-PNI foci.

Ligand–Receptor analysis

The GeoMx DSP platform evaluates gene expression from hundreds of cells in morphological areas of interest, thus producing localized bulk RNAseq data. We used the BulkSignalR [24] (version 0.0.9) R package to explore the expression patterns of ligands, receptors and downstream pathways in the cancer and nerve compartments collected from PNI and non-PNI foci. Normalized/batch-corrected data from standR were used as input for BulkSignalR. Ligand–receptor pathway correlation was considered significantly different between PNI and non-PNI if the adjusted *p*-value was <0.20 (Benjamini–Hochberg).

## 3. Results

We compared the gene expression of cancer and nerves in PNI vs. non-PNI foci in PDAC patient samples. Figure 1a shows a representative tumor specimen, with examples of morphology (adjacent H&E—Figure 1b–d) and ROIs/AOIs (GeoMx—Figure 1f,h,j) selected for gene expression analysis. A PNI region where cancer cells invaded the layers of nerve fibers, from which gene expression was measured in ROIs/AOIs from the cancer and nerve compartments is presented in Figure 1b,e,f. For non-PNI, the cancer compartment’s AOIs were selected away from any visible nerves (Figure 1c,g,h), and for the nerve compartment, ROIs/AOIs were selected in nerves with no visual signs of invasion and located outside of the TME (Figure 1d,i,j). A total of 151 ROIs/AOIs were collected. However, 77 were discarded due to excessive variability in the number of nuclei or library size or for other technical issues (Appendix A).

A principal component analysis (PCA) of 74 AOIs/ROIs from 10 patients/11 slides showed that gene expression counts separated nerves and cancer compartments (Figure 2a) despite the expression variability between patients/slides (Figure 2b). However, as our study’s aim was to evaluate the expression profile of cancer and nerve compartments located in PNI vs. non-PNI foci, the expression counts of nerve and cancer compartments were analyzed separately.

For the nerve compartment, 17 full ROIs (PGP9.5-positive) from six patients were used. The ROIs that passed QC were no longer from matched PNI and non-PNI foci within each patient/slide (Appendix A). In this analysis, patient/slide was used in the batch correction with the resulting weight matrices used as co-variates in a linear model. PCA showed the separation of the expression of the nerve ROIs collected in PNI and non-PNI foci (Figure 2c,d).

For the cancer compartment, 40 AOIs segmented by PanCK positivity across eight patients passed QC and had AOIs in both PNI and non-PNI foci within a slide. These were used for downstream analysis (Appendix A). A PCA of the expression from cancer AOIs (Figure 2e) showed that there were differences between PNI and non-PNI within patients/slides, but there was a greater variability in expression between patients (Figure 2f) as expected due to tumor heterogeneity. For differential gene expression analysis, patients were included as a co-variate in a linear model to identify similar differences between PNI and non-PNI foci across patients.

### 3.1. Differential Gene Expression and Pathway Analyses of Cancer Compartments in PNI vs. Non-PNI Foci

An analysis of cancer compartments in PNI vs. non-PNI foci identified 1554 genes as differentially expressed (FDR < 0.20 and FC > 1.4), with 288 genes up-regulated and 1266 down-regulated in the PNI cancer compartment when compared to non-PNI (Figure 3a and Appendix A). The high number of down-regulated genes could reflect the smaller number of cells in PNI AOIs. Despite including only AOIs with a reduced range of the number of nuclei (200 to 700) in the analysis, the number of nuclei/AOIs and library sizes from PNI foci were smaller than those from non-PNI (Appendix A). We therefore only explored pathways significantly enriched using up-regulated genes in PNI foci (Figure 3b and Appendix A).

The genes AKT Serine–Threonine Kinase 1 (AKT1) and SRC Proto-Oncogene, Non-Receptor Tyrosine Kinase (SRC), were up-regulated in the cancer compartment of PNI foci compared to non-PNI. These genes are known to participate in a diverse range of cancer-related pathways, including the epithelial-to-mesenchymal transition, invasion, tumor endothelial barrier disruption, cancer metastasis, gene transcription, immune response, cell adhesion, cell cycle progression, apoptosis and migration.

The genes up-regulated in the cancer compartment in PNI predicted the activation of cell migration, movement of fibroblasts, outgrowth of axons and up-regulation of Hepatocyte Growth Factor (HGF) (Figure 3b). HGF/c-Met signaling has been shown to increase cancer cell invasion and dorsal root ganglia outgrowth in vitro [25]. Pathways significantly (*p* ≤ 0.05) enriched in the cancer cells invading the nerves (Appendix A) also included ERK5 signaling, signaling by NTRK1, Gap Junction signaling, the HOTAIR Regulatory Pathway and the WNT/SHH Axonal Guidance Signaling Pathway, all previously reported to be involved in PDAC carcinogenesis and/or PNI [10,26,27,28].

Pathway analysis predicted IL4 activation (Figure 3b) in the cancer compartment in PNI. IL4 is a key cytokine for type 2 immune response and has been suggested to play a role in nerve injury and regeneration [29]. Pathways related to neuronal biology, such as Myelination Signaling, Reelin Signaling in Neurons, Endocannabinoid Developing Neuron Pathway and Opioid Signaling Pathway (Appendix A), were also significantly enriched in the cancer compartment of PNI foci. These results are supported by recent studies using the single-cell nucleus sequencing and laser micro-dissected RNASeq of PDAC that identified the subsets of cancer cells in the TME with a neuronal-like expression profile [19,21], and the authors suggested a potential role of these cancer cells in PNI.

A key member of the Endocannabinoid Developing Neuron Pathway is Monoacylglycerol lipase (MGLL). MGLL hydrolyzes 2-arachidonoylglycerol (2-AG), an endogenous agonist of the cannabinoid receptors 1 and 2 [30], which are involved in the control of the perception of pain. The MGLL gene is highly expressed in pancreatic cancer compared to the normal pancreas (Appendix A) [31]. Our results showed that MGLL gene (Figure 3c) and protein expression levels (Figure 4a,b,c,d) are higher in the cancer compartment within PNI compared to non-PNI regions. However, we observed a high heterogeneity in MGLL immunoreactivity in cancer cells in the TME (Figure 4b,c,d, Appendix A). This heterogeneity in cancer cells’ transcriptional program has been previously described [21] in micro-dissected cancer cells from two distinct histological features, which were associated with four transcriptional programs that coexist in PDAC tumors and may be driven by microenvironmental signals interacting with cancer cells. We investigated MGLL expression in the data associated with this distinct transcriptional program [21] (Appendix A). MGLL expression was higher in cancer cells with the transcriptional program termed glandular, which was enriched in PNI foci [21].

Another pathway identified to be enriched in the cancer compartment in PNI but not previously reported as associated with PNI was polyamine regulation (Figure 3b and Appendix A), which includes the gene Spermidine–Spermine N1-Acetyltransferase 1 (SAT1), which was up-regulated in cancer cells invading nerves (Figure 3d). SAT1 catalyzes the N(1)-acetylation of spermidine and spermine, which can then either be excreted from the cell or serve as substrates for polyamine oxidase (PAOX) [32]. SAT1 has a higher expression in pancreatic cancer compared to the normal pancreas (Appendix A) [31]. We also investigated SAT1 expression in data [21] from micro-dissected cancer cells associated with four transcriptional programs. Cancer cells with glandular and transitional transcriptional programs reported to be associated with PNI [21] had higher expression levels of SAT1 (Appendix A).

### 3.2. Differential Gene Expression and Pathway Analyses of Nerves with PNI vs. Non-PNI Evidence

A differential gene expression analysis of nerves was performed using 17 ROIs (PGP9.5-positive) collected from six patients (Appendix A). As PGP9.5 is also a marker for neuroendocrine cells, all PGP9.5-positive ROIs were morphologically confirmed as nerves in adjacent H&E sections (Figure 1d). Nerve fibers are mainly composed of layers of connective tissue, axons and Schwann cells, but they also contain immune cells, fibroblasts, blood and lymphatic vessels. Targeted ROIs would potentially collect tags for expressed genes from all cell types present in the nerve fibers. The sizes of ROIs collected from nerves with PNI were smaller than those of nerves with non-PNI evidence, despite including only ROIs with > 125 nuclei in the analysis (Appendix A). When comparing the gene expression of nerves with PNI vs. without visible invasion (non-PNI), 338 genes were differentially expressed (FDR < 0.20 and FC > 1.4), with 178 genes up-regulated and 160 down-regulated in nerves with PNI compared to non-PNI (Figure 5a, Appendix A).

Ribosomal genes and pathways involved in translation regulation were enriched in nerves with PNI (Appendix A). These findings agreed with previous reports that mRNA localization and the subsequent translation in peripheral neurons are regulated by external stimuli, which leads to rapid local protein translation to support axon guidance, neuron migration, synaptic activity and regeneration after nerve injury [33].

Analysis using up-regulated genes in nerves with PNI identified enrichment in the pathways involved in the regulation of the immune system, such as Neutrophil degranulation, Leukocyte extravasation signaling, Response to elevated platelet cytosolic Ca2+ and IL-10 Signaling (Appendix A), and predicted the activation of tumor necrosis factor (TNF), interleukin 6 (IL6), interleukin 1 beta (IL1B), interleukin 1 alpha (IL1A), Interferon gamma (INFG) and Transforming Growth Factor Beta 1 (TGFB) (Figure 5b).

Pathway analysis also predicted the activation of the migration, binding and adhesion of cancer cells (Figure 5b), reinforcing the potential role of nerves in the crosstalk with cancer cells and supporting that nerves are not bystanders of cancer cell invasion. Pathways reported to be involved in PNI and PDAC tumorigenesis such as Axonal Guidance and the ROBO-SLIT Signaling Pathway were also enriched in nerves with PNI (Appendix A).

The Neuron Growth-Associated Protein 43 (GAP43) and Nestin (NES) genes were within the top up-regulated genes in nerves with PNI compared to nerves without PNI evidence (Appendix A, Figure 5 c,d). Nestin and GAP43 have very low to no expression in PDAC or the normal pancreas (Appendix A). High GAP43 expression has been reported in nerves at the site of injury during the early stages of nerve regeneration [34] and nerve fibers in PDAC and chronic pancreatitis compared to normal pancreas [4].

Here, Nestin gene expression and immunoreactivity in IHC were higher in nerves with PNI compared with nerves without signs of invasion (non-PNI) (Figure 6a–d Appendix A). Nestin is a marker of stem/progenitor cells in different tissues. In adult tissues, Nestin expression was reported in tissue regeneration and healing in response to injury and revascularization [35]. Our results agreed with previous reports that nerves invaded by cancer cells or located in the pancreatitis lesions adjacent to the cancer cells were Nestin-positive, whereas nerves without visible signs of invasion located outside the TME [36] or in the normal pancreas [35] were Nestin-negative. These studies suggested that Nestin may play a role in neural remodeling and interactions with cancer cells.

### 3.3. Receptor–Ligand Analysis

Cell-to-cell crosstalk through receptor–ligand interactions is key in tumor progression and TME homeostasis. Unlike single-cell technologies, the GeoMx platform measures gene expression as an average of hundreds of cells, which reduces the cellular complexity when compared to bulk RNASeq but limits the analysis of receptor–ligand interactions in a cell-to-cell context. Here, we used BulkSignalR to assess the gene expression patterns of ligands, receptors and targeted genes in downstream pathways (LRP). We compared LRP expression patterns in the cancer and nerve compartments in PNI vs. non-PNI foci.

The expression patterns of four LRPs were significantly different between PNI and non-PNI cancer compartments (Appendix A). EPHA2/EFNA1 had a positive correlation in PNI and a negative correlation in non-PNI cancer compartments (Figure 7a). The best inferred pathway involving the EPHA2 receptor was EPHA-mediated growth cone collapse (Appendix A). Eph receptors and Ephrins are both membrane-anchored, with their interactions impacting pathways such as cell adhesion, cell migration and axon extension or repulsion [37]. A previous study has reported a correlation between EPHA2/EFNA1 expression and perineural and vascular invasion in Adenoid Cystic Carcinoma [38].

In the nerve compartment, 17 LRPs had different expression patterns in PNI compared to non-PNI foci (Appendix A). Most significant interactions involved integrins (ITGAV and ITGB1) linked to the pathways’ L1CAM interactions and Basigin interactions (Appendix A). Previous studies suggest that integrins and L1CAM are involved in PDAC tumorigenesis [39,40,41]. It has also been reported that integrins play a role in axonal regeneration and repair after nerve injury [42]. Less is known about the involvement of receptor–ligand BMPR2/GDF7 (Figure 7b) and CD47/THBS2 (Figure 7c) in PNI. THBS2 knockdown has been shown to decrease the proliferation, migration and invasion of pancreatic cancer cells in vitro [43]; however, its role in the nerves and PNI needs further investigation.

## 4. Discussion

Pancreatic adenocarcinoma (PDAC) remains a lethal disease with limited treatment successes, despite the improvements in surgical techniques and adjuvant and neoadjuvant therapies for treating PDAC in recent years [44]. With the increase in disease incidence [2,45], it is key that we further understand the molecular interactions of this cancer to identify more effective therapies and/or improve the quality of life. Perineural invasion (PNI) is a common feature in PDAC, and it is associated with a poor prognosis, metastatic spread [13] and pain severity [4].

In the past two decades, our understanding of PNI has increased, with most of this knowledge coming through pre-clinical in vitro or in vivo studies. Despite these studies being key to our understanding, they may not fully represent the plasticity of PNI mechanisms and the complexity of the TME. Further exploration using new technologies such as spatial transcriptomics to further untangle the complexity of the PNI microenvironment in patients’ samples may provide new insights into this common pathological feature of PDAC that contributes to cancer pain, recurrences and poorer prognosis for patients.

Here, we showed that spatial transcriptomics can be used with PDAC archival samples for discovery and hypothesis generation. Future studies would benefit from a more careful collection of tags from ROIs/AOIs with a closer matching of the sizes and number of nuclei across PNI and non-PNI foci to improve discoveries. In our experience with archival PDAC samples, which were 10 to 12 years old, ROIs/AOIs required more than 120 nuclei to generate consistent library sizes. Despite the limitations we faced, we identified novel genes and pathways potentially involved in PNI, together with previously reported genes and pathways involved in PNI.

Our study suggests that both cancer and nerve compartments may contribute to immune regulation and tissue remodeling in the PNI TME. The activation of *IL4* expression in the cancer compartment in PNI is supported by a previous report that the PNI microenvironment was associated with decreased CD8+ T and Th1 cells and increased Th2 cells [15]. *IL4* is a key cytokine for type 2 immune response and for wound healing during nerve injury, leading to neural remodeling and Schwann cell differentiation to facilitate axon growth [29]. In nerves with PNI, pathway analyses predicted the activation of *TNF*, *IL6* and *IL1B* compared to nerves with non-PNI evidence. These pro-inflammatory cytokines have been suggested to cause neuronal sensitization by stimulating specific receptors expressed in the nociceptive neurons [46]. Further investigations are needed to confirm whether the PNI environment can release signals to recruit macrophages and induce the polarization of tumor-associated macrophages. A previous study showed more macrophages in the perineural microenvironment of PNI in PDAC than non-PNI [47]. Altogether, these results exemplify the complexity of immune regulation in the TME and support a previous report [15] that states that PNI may contribute to an immune-suppressed microenvironment in PDAC.

Our study showed the up-regulation of *GAP43* expression in nerves with PNI. *GAP43* has been associated with neuronal plasticity and shown to have increased expression in nerves in PDAC and pancreatitis [4]. *GAP43* is expressed by Schwann cell precursors and non-myelinating Schwann cells; however, in injured nerves, myelinating Schwann cells can also express *GAP43* [48], which may account for its up-regulation in our results. Further, Nestin expression and immunoreactivity were also up-regulated in nerves with PNI when compared with nerves without evidence of invasion, confirming previous reports that showed higher Nestin expression in nerves with PNI than that in non-PNI nerves [35,36]. Nestin expression is increased in the brain and spinal cord in response to nerve injury and may contribute to the activation of progenitor cells after injury [49]. A proteomic analysis of micro-dissected nerves with PNI and non-PNI in PDAC showed the up-regulation of Nestin and GAP43 protein levels [50] in nerves with PNI compared to nerves without PNI. Together, the up-regulation of *GAP43* and Nestin in nerves with PNI suggests similarities between PNI and regeneration after nerve injury. These findings are supported by previous studies that showed similarities between PNI and nerve regeneration after injury in in vivo models [18,51] and oral cancers [17].

The gene expression of cancer cells (PanCK-positive) suggests a potential role for polyamine metabolism in PNI. *SAT1*, a gene that encodes a rate-limiting enzyme in polyamine metabolism, was up-regulated in the cancer compartment of PNI foci. A recent study [52] showed that stroma-derived acetate enabled cancer cell survival under acidosis via the ACSS2–SP1–SAT1 axis, and its inhibition diminished the tumor burden in a mouse model. The concentrations of putrescine and cadaverine have been reported to be significantly higher in pancreatic cancer, with similar levels for spermidine and spermine compared to the normal pancreas. N1-acetylspermidine catalyzed by the SAT1 enzyme was only detectable in cancer tissue [53]. The inhibition and overexpression of *SAT1* in vitro reduced and increased, respectively, the levels of intra- and extracellular N1-acetylspermidine [54]. Public data showed that PDAC had a higher expression of *SAT1* compared to the normal pancreas, and here, our data suggested a higher expression of *SAT1* in cancer cells in PNI than that in non-PNI foci. In cancer cells, increased intracellular polyamine levels enhance malignancy potential and decrease anti-tumor immunity [55], whereas in nerves, endogenously produced polyamines could play a role in inflammatory pain [56]. A study showed that the treatment of neuronal cells with spermidine increased the proportion of cells with neurite growth [57]. We postulate that SAT1-mediated polyamine levels may also play a role in cancer growth and PNI. The exact mechanisms that regulate the SAT1-mediated acetylation of spermidine or spermine in PDAC and its role in PNI and cancer progression have yet to be fully understood. Further studies using patient-derived models [58] are needed to establish mechanisms that drive the expression of *SAT1* in PDAC and clarify its role in tumorigenesis and PNI.

The Endocannabinoid Developing Neuron pathway was enriched in the cancer cells invading the nerves. In this pathway, MGLL catalyzes the hydrolysis of 2-arachidonoylglycerol (2-AG), an antagonist of the endocannabinoid system, which plays a crucial role in pathophysiological processes such as cancer proliferation, inflammation, pain and neuroprotection [59,60]. MGLL hydrolyzes 2-AG into arachidonic acid and glycerol, which may contribute to inflammation and the signaling of pro-tumorigenic signaling, respectively [30]. We hypothesize that the up-regulation of *MGLL* leads to the hydrolysis of 2-AG increasing lipid signaling, which may play a role in PDAC growth, migration and invasion potential and may contribute to cancer pain. MGLL inhibitors have been evaluated in models of osteoarthritis [61], bone cancer pain [62] and neuropathic pain [63], including pain induced by chemotherapeutic drugs [64,65]. These studies showed that MGLL inhibition increased the levels of 2-AG and alleviated pain, hypersensitivity and allodynia. 2-AG has also been shown to inhibit pancreatic cancer cell proliferation in vitro and promote the expansion of myeloid-derived suppressor cells in vivo [66]. A recent study showed that pain intensity was inversely associated with 2-AG blood levels in patients with pancreatitis [67]. Another study did not associate MGLL protein expression in nerves with pain levels in pancreatic cancer [68]; however, this study did not evaluate whether the expression of MGLL in cancer cells could be associated with pain experienced by patients. The inhibition of MGLL reduced tumor growth in a mouse model of prostate cancer [69], endometrial cancer cells [70] and primary pancreatic cancer cultures [71]. It has been shown that the overexpression of *MGLL* promotes tumor migration, invasion and proliferation [72]. Public data showed that *MGLL* expression was higher in PDAC compared to the normal pancreas, and our data suggested that *MGLL* gene expression and immunoreactivity were higher in PNI foci, potentially indicating that cancer cells are more proliferative and invasive in these regions. However, there is a high heterogeneity in MGLL expression in the TME that needs to be better understood. We speculate that the high levels of MGLL in PDAC play a role in PNI and tumorigenesis and could be associated with the pain experienced by patients, indicating the need to further investigate the role of MGLL in PDAC and to determine whether direct targeting could be an alternative treatment for this dismal disease.

## 5. Conclusions

This exploratory study shows that spatial transcriptomics allows for the investigation of patient samples for hypothesis generation and the discovery of novel mechanisms potentially involved in PNI and PDAC carcinogenesis. Here, we identified two potential pathways, polyamine and endocannabinoid metabolism, with key members in these pathways, the *SAT1* and *MGLL* genes, having higher expression in PDAC compared to the normal pancreas with an up-regulation in PNI compared with non-PNI foci. These genes were targeted in vitro and could potentially impact cancer growth and cancer pain. Further investigations of our findings could lead to future treatment options and pain management strategies for the debilitating pain experienced by PDAC patients and could help improve their quality of life.

## Figures and Tables

**Figure 1 cancers-17-00852-f001:**
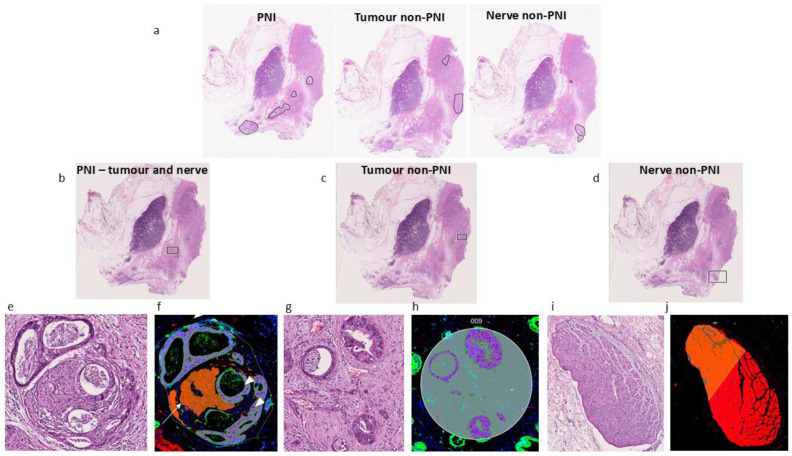
Representative images of regions collected in this study. (**a**) H&E image of sample used in this study with circled regions containing PNI and cancer or nerves (non-PNI). (**b**–**d**) Black squares in specimen indicate regions selected for GeoMx. (**e**) Zoomed image of region selected in H&E in (**b**) (PNI region). (**f**) Same region presented in (**e**), showing GeoMx DSP ROI and segmented AOIs in terms of PanCK and PGP9.5 positivity for collection of tags. Cancer (arrowhead) and nerve (arrow) compartments in PNI focus. (**g**) H&E of cancer compartment (non-PNI) indicated in (**c**) (square), cancer away from any visible nerve. (**h**) GeoMx DSP ROI showing segmented AOI in terms of PanCK positivity (purple) for collection of tags. (**i**) H&E image of uninvolved nerve (non-PNI) away from tumor microenvironment indicated in (**d**) (square), with no visible cancer invasion. (**j**) GeoMx DSP of collected ROI. Regions (**e**–**j**) are examples of regions where gene expression was measured; selection of regions was based on visual inspection of adjacent H&E and PanCK or PGP9.5 positivity for collection of expression tags.

**Figure 2 cancers-17-00852-f002:**
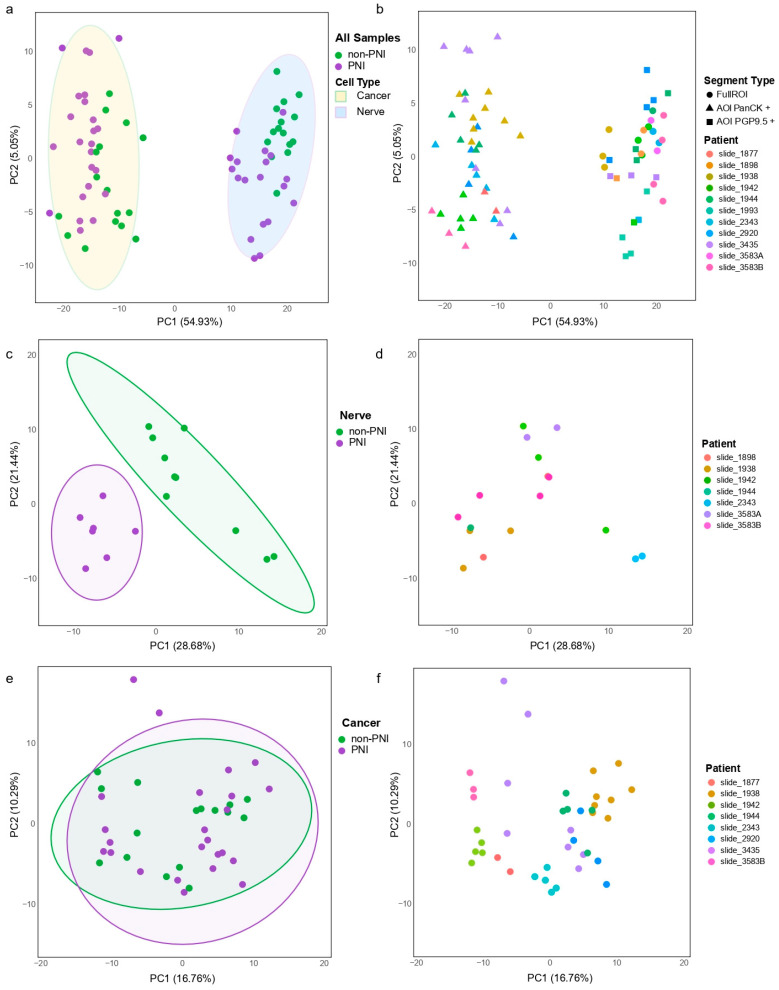
Expression measured in 74 regions collected by DSP GeoMx. (**a**) Principal component analysis (PCA) of normalized expression of regions profiled in cancer and nerve compartments that pass QC (n = 74). Gene counts showed separations of cancer and nerve regions. Each dot represents expression of region profiled and is colored by location in PNI or non-PNI foci. Circles indicate 95th percentile of expression. (**b**) PCA, same data presented in (**a**), colored by patient from whom samples were collected. (**c**) Normalized expression of full ROIs collected from nerve compartments (n = 17) in PNI or non-PNI foci. PCA separated nerves with PNI from non-PNI (without visual signs of invasion). (**d**) Same data presented in (**c**), colored by patient from whom samples were collected. (**e**) Normalized expression of AOIs (areas segmented by PanCK positivity) collected from cancer compartments (n = 40) in PNI or non-PNI foci. (**f**) Same data presented in (**e**), colored by patient from whom samples were collected.

**Figure 3 cancers-17-00852-f003:**
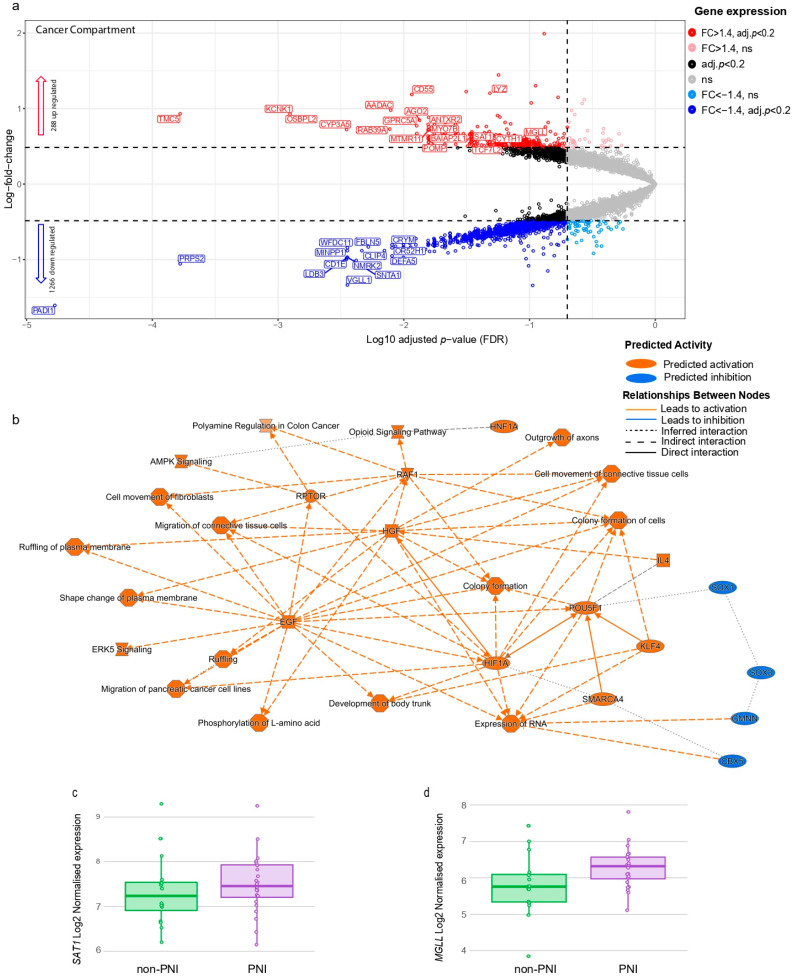
Differentially expressed genes and pathways enriched in cancer compartment of PNI foci. (**a**) Volcano plot showing differentially expressed genes in PNI cancer compartment compared with non-PNI foci. Each dot represents gene, with red dots indicating genes significantly up-regulated (adjusted *p*-value < 0.20 and fold change > 1.4), dark blue dots indicating genes that are significantly down-regulated (adjusted *p*-value < 0.20 and fold change < −1.4) and black dots indicating genes differentially expressed (adjusted *p*-value < 0.20) with fold change between −1.4 and 1.4. Pink and light blue dots are genes with fold change >1.4 and fold change < −1.4, respectively, but are not statistically significant (adjusted *p*-value > 0.20). Dotted lines represent threshold for statistical significance (adjusted *p*-value < 0.20) and threshold for fold change (fold change > 1.4 or <−1.4). Gene symbols for some significantly differentially expressed genes are shown (for complete list, see Appendix A). (**b**) Pathways predicted to be activated or inhibited by up-regulated genes in PNI cancer compartment (n = 288), (for full list of pathways, see Appendix A). Pathways were obtained using IPA and up-regulated genes. (**c**) MGLL gene expression in cancer AOIs (PanCK-positive) in PNI and non-PNI regions (adjusted *p* = 0.091). (**d**) SAT1 gene expression in cancer AOIs (PanCK-positive) in PNI and non-PNI regions (adjusted *p* = 0.035).

**Figure 4 cancers-17-00852-f004:**
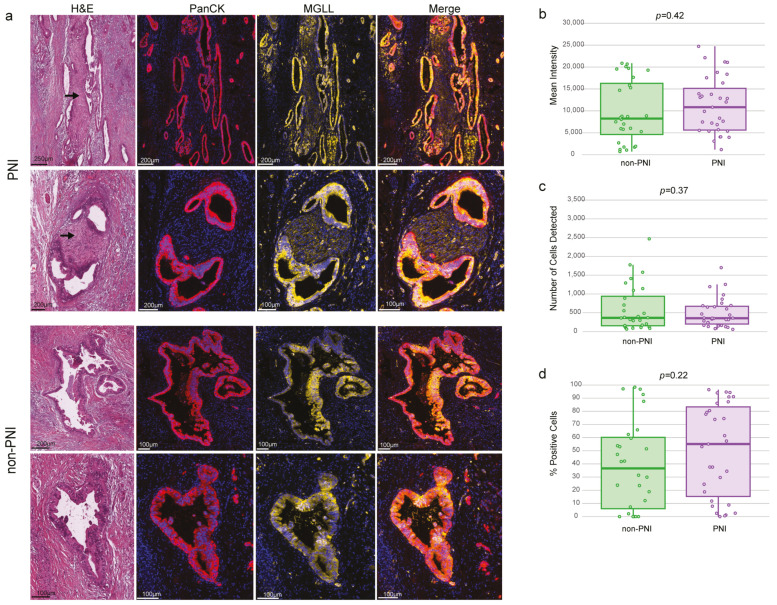
Immunohistochemistry (IHC) for MGLL. (**a**) Representative histology (H&E staining) images of regions selected to measure immunoreactivity of MGLL in cancer cells in PNI and non-PNI foci. Arrow indicates nerve. IHC double immunofluorescence of morphological regions selected with PanCK and MGLL staining. (**b**) Mean intensity of MGLL in PanCK-positive regions (PNI and non-PNI foci). (**c**) Number of cells detected/region. (**d**) Percentage of cells with positive expression of MGLL. IHC was performed in 8 PDAC samples and quantification of MGLL immunoreactivity performed in 59 regions (31 PNI and 28 non-PNI). Indicated *p*-values from T-test.

**Figure 5 cancers-17-00852-f005:**
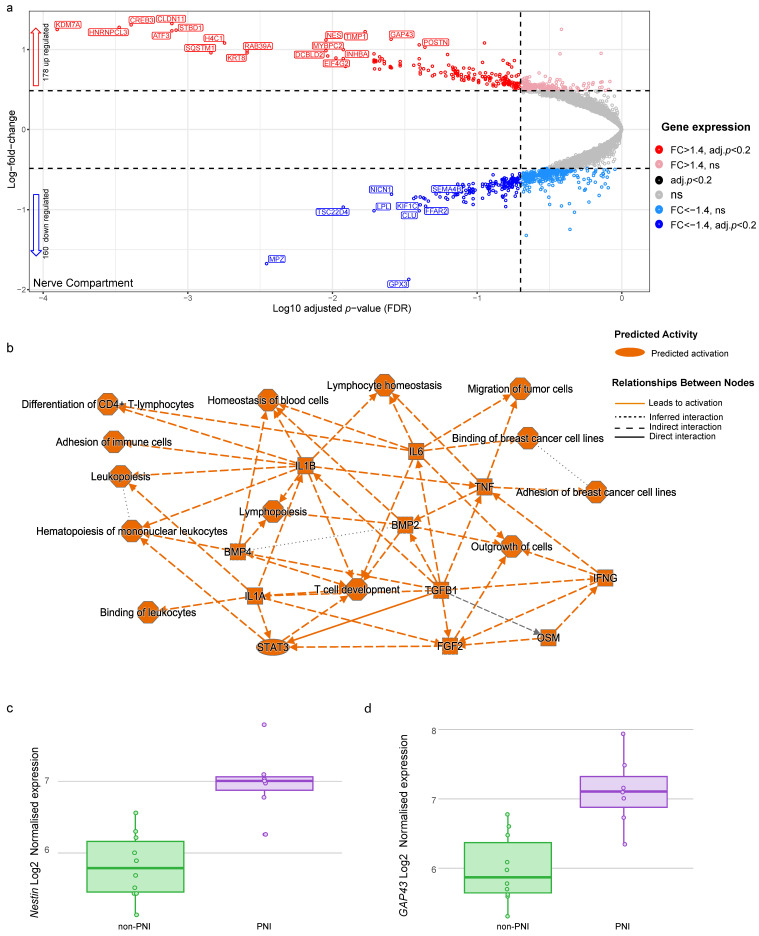
Differentially expressed genes and pathways enriched in nerve compartment of PNI foci. (**a**) Volcano plot showing differentially expressed genes in PNI nerve compartment compared with non-PNI foci. Each dot represents gene, with red dots indicating genes that are significantly up-regulated (adjusted *p*-value < 0.20 and fold change > 1.4), dark blue dots indicating genes that are significantly down-regulated (adjusted *p*-value < 0.20 and fold change < −1.4) and black dots indicating genes that are significantly differentially expressed (adjusted *p*-value < 0.20) with fold change between −1.4 and 1.4. Pink and light blue dots are genes with fold change > 1.4 and fold change < −1.4, respectively, but are not statistically significant (adjusted *p*-value > 0.20). Dotted lines represent threshold for statistical significance (adjusted *p*-value < 0.20) and threshold for fold change (fold change > 1.4 or <−1.4). Gene symbols for significantly differentially expressed genes are shown (for complete list, see Appendix A). (**b**) Pathways predicted to be activated by up-regulated genes (n = 178) (for full list of pathways, see Appendix A). Pathways were obtained using IPA. (**c**) Normalized gene expression of Nestin (NES) (adjusted *p* = 0.009). (**d**) GAP43’s normalized expression in nerves with PNI and non-PNI evidence (adjusted *p* = 0.025).

**Figure 6 cancers-17-00852-f006:**
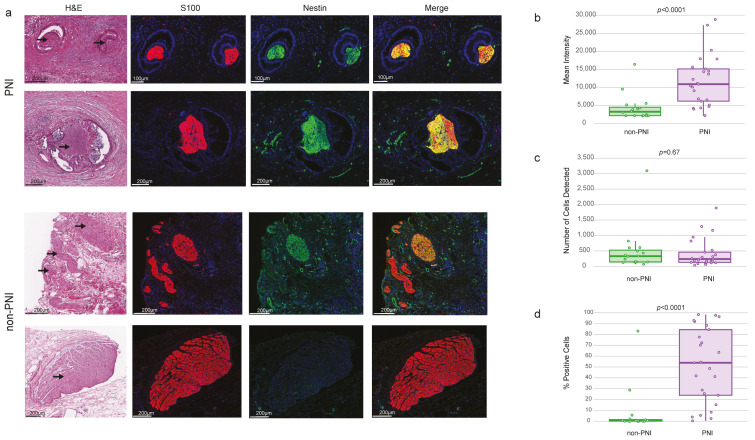
Immunohistochemistry (IHC) of Nestin. (**a**) Representative histology (H&E staining) images of regions selected to measure immunoreactivity of Nestin in nerves with PNI and non-PNI foci. Arrow indicates nerve fibers. IHC double immunofluorescence of morphological regions selected in H&E with Nestin and S100 staining. (**b**) Mean intensity of Nestin in S100-positive regions in nerve areas in PNI and non-PNI foci. (**c**) Number of cells detected/region. (**d**) Percentage of cells with positive expression of Nestin. IHC was performed in 7 PDAC samples and quantification performed in 41 regions (25 PNI and 16 non-PNI). Indicated *p*-values from T-test.

**Figure 7 cancers-17-00852-f007:**
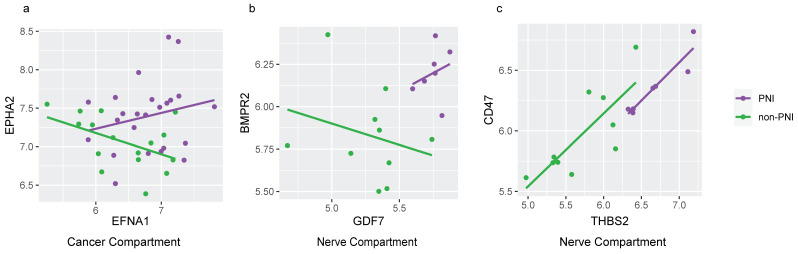
Receptor–ligand expression in cancer and nerve compartments in PNI and non-PNI foci. Scatterplots of normalized gene expression of significant differentially expressed ligand–receptor pathways (LPRs), obtained using BulkSignalR (adjusted *p* value < 0.20) with superimposed linear model line. (**a**) EPHA2 (receptor) and EFNA1 (ligand) expression in cancer compartment of PNI (purple) and non-PNI (green). (**b**) BMPR2 (receptor) and GDF7 (ligand) expression in nerve compartment of PNI (purple) and non-PNI (green). (**c**) CD47 (receptor) and THBS2 (ligand) expression in nerve compartment of PNI (purple) and non-PNI (green).

## Data Availability

Spatial transcriptomic raw counts are presented in Appendix A.

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
