# Peer review of "Spatial Transcriptomics Reveals Novel Mechanisms Involved in Perineural Invasion in Pancreatic Ductal Adenocarcinomas"

_cancers, 2025, doi:10.3390/cancers17050852_

Round 1
Reviewer 1 Report
Comments and Suggestions for Authors
Dear authors and editors!
The paper attempts to identify novel genes regulating PDAC perineural invasion. Authors suggest that “Endocannabinoid and Polyamine metabolism may contribute to PNI, cancer growth and cancer pain.” and that targeting these pathways might be beneficial for cancer management. Overall, the paper is of sufficient quality, although I have a few minor questions.
-
Lines 134-140. Please describe in more details the difference between AOI and ROI. Apparently this distinction does not appear further in the text, I am not sure why you need it?
-
Figures 3C and 3D do not show p-values and something is wrong with the axis labels.
-
Figure 4D. First of all, you agree of course that IHC is not that quantitative. Especially if the changes if gene expression (Figure 3C) are minor. Have you tried to do a paired comparison, from the same patient of perineural v.s. non-perineural regions? If such comparison is possible.
-
Line 431, Receptor-ligand analysis. In addition, please combine both cancer and neural datasets together to identify potential molecules that might contribute to the crosstalk between the two.
Reviewer 2 Report
Comments and Suggestions for Authors
This study investigates novel mechanisms underlying perineural invasion (PNI) in pancreatic ductal adenocarcinoma (PDAC). PNI is characterized by the infiltration of cancer cells into pancreatic nerves and is associated with poor prognosis, local recurrence, and severe cancer-related pain. Using spatial transcriptomics, the study analyzed the transcriptomic profiles of cancer and nerve cells from both PNI and non-PNI regions in tissue samples from 13 PDAC patients. The analysis revealed several known pathways involved in PNI, including axon guidance and ROBO-SLIT signaling, while also highlighting the influence of PNI on the tumor microenvironment's immune landscape. Furthermore, the study uncovered potential roles of endocannabinoid signaling and polyamine metabolism in promoting PNI, cancer progression, and pain development. These findings provide valuable insights for developing new therapeutic strategies and pain management approaches in PDAC. In my assessment, this manuscript demonstrates significant merit and innovation, with the scope of work aligning well with the journal's requirements. It is recommended for acceptance following minor revisions. The specific revision suggestions are outlined below.
1. The author should allocate additional space to elaborate on the pertinent background of pancreatic cancer, thereby enabling readers to grasp the current research status of this disease more swiftly and comprehensively.
2. If the author intends to utilize a database in the path analysis, it is imperative to specify which database is employed and provide the corresponding link.
3. The author should meticulously refine the discussion section to guarantee logical coherence and rigor. Additionally, incorporating a broader range of references will enhance the scientific validity and credibility of the work.
4. The author should meticulously review the grammar and tense throughout the document to ensure its accuracy.
5. It is recommended to cite the following literature:
[1] A. Gu, J. Li, S. Qiu, S. Hao, Z.-Y. Yue, S. Zhai, M.-Y. Li, Y. Liu, Pancreatic cancer environment: From patient-derived models to single-cell omics. Mol. Omics 2024, 20, 220–233. DOI: 10.1039/D3MO00250K
[2] Wang J, Liao Z-X. Research progress of microrobots in tumor drug delivery. Food & Medicine Homology, 2024, 1(2): 9420025. https://doi.org/10.26599/FMH.2024.9420025
[3] X. Wang, K. Mao, X. Zhang, Y. Zhang, Y.-G. Yang, T. Sun, Red blood cell derived nanocarrier drug delivery system: A promising strategy for tumor therapy. Interdiscip. Med. 2024, 2, e20240014. https://doi.org/10.1002/INMD.20240014
Comments on the Quality of English LanguageThe English could be improved to more clearly express the research.
Author Response
Reviewer 2:
This study investigates novel mechanisms underlying perineural invasion (PNI) in pancreatic ductal adenocarcinoma (PDAC). PNI is characterized by the infiltration of cancer cells into pancreatic nerves and is associated with poor prognosis, local recurrence, and severe cancer-related pain. Using spatial transcriptomics, the study analyzed the transcriptomic profiles of cancer and nerve cells from both PNI and non-PNI regions in tissue samples from 13 PDAC patients. The analysis revealed several known pathways involved in PNI, including axon guidance and ROBO-SLIT signaling, while also highlighting the influence of PNI on the tumor microenvironment's immune landscape. Furthermore, the study uncovered potential roles of endocannabinoid signaling and polyamine metabolism in promoting PNI, cancer progression, and pain development. These findings provide valuable insights for developing new therapeutic strategies and pain management approaches in PDAC. In my assessment, this manuscript demonstrates significant merit and innovation, with the scope of work aligning well with the journal's requirements. It is recommended for acceptance following minor revisions. The specific revision suggestions are outlined below.
- The author should allocate additional space to elaborate on the pertinent background of pancreatic cancer, thereby enabling readers to grasp the current research status of this disease more swiftly and comprehensively.
Response: We added some facts about the disease in the introduction and Discussion, however due to the limitations to the length of the manuscript we kept to a brief description.
Added: Lines 52 to 56: Pancreatic cancer incidence is rising at a rate of 0.5% to 1.0% per year and it is predicted to become the second-leading cause of cancer-related mortality by 2030 [2]. Currently, available therapies have only modest effect on patients’ outcomes [2]. These facts highlight the urgency to better understand the PDAC microenvironment to identify novel treatments for this deadly disease.
Discussion Lines 482-488): Pancreatic adenocarcinoma (PDAC) remains a lethal disease with limited treatment successes, despite the improvements in surgical techniques, adjuvant and neoadjuvant therapies for treating PDAC in recent years [44]. With the increase of disease incidence [45], it is key that we further understand the molecular interactions of this cancer to identify more effective therapies and/or improve quality of life.
- If the author intends to utilize a database in the path analysis, it is imperative to specify which database is employed and provide the corresponding link.
Response: It was indicated in the Material and Methods Line 176: We use QIAGEN Ingenuity Pathway Analysis, a commercial platform that uses proprietary curated knowledge database.
- The author should meticulously refine the discussion section to guarantee logical coherence and rigor. Additionally, incorporating a broader range of references will enhance the scientific validity and credibility of the work.
Response: We have reviewed the discussion and incorporated new references
- The author should meticulously review the grammar and tense throughout the document to ensure its accuracy.
Response: We have reviewed the grammar and tense throughout the document.
- It is recommended to cite the following literature:
[1] A. Gu, J. Li, S. Qiu, S. Hao, Z.-Y. Yue, S. Zhai, M.-Y. Li, Y. Liu, Pancreatic cancer environment: From patient-derived models to single-cell omics. Mol. Omics 2024, 20, 220–233. DOI: 10.1039/D3MO00250K
[2] Wang J, Liao Z-X. Research progress of microrobots in tumor drug delivery. Food & Medicine Homology, 2024, 1(2): 9420025. https://doi.org/10.26599/FMH.2024.9420025
[3] X. Wang, K. Mao, X. Zhang, Y. Zhang, Y.-G. Yang, T. Sun, Red blood cell derived nanocarrier drug delivery system: A promising strategy for tumor therapy. Interdiscip. Med. 2024, 2, e20240014. https://doi.org/10.1002/INMD.20240014.
Response: We have included reference Gu et al. 2024. Line 555: Further studies using patient-derived models [58] are needed to establish mechanisms that drive expression of SAT1 in PDAC and clarify its role in tumorigenesis and PNI.

Reviewer 3 Report
Comments and Suggestions for Authors
The paper "Spatial transcriptomics reveals novel mechanisms involved in perineural invasion in pancreatic ductal adenocarcinomas" is a well-conducted study that employs a range of state-of-the-art instrumental techniques. As a result, all findings are well supported by the experimental data. I have one recommendation regarding the organization of the manuscript: Given the numerous abbreviations used in the paper, I suggest creating a separate "Abbreviations" section. This would help readers understand and navigate the paper more easily.
And few minor comments:
Line 174: Leica (?).
Lines 177, 182: p.H., probably pH (?)
Line 213: of interest (!)
Fig. 1, b. No arrows and arrowheads are seen.
Fig. S6, S7. Please, describe, what are the straight lines, how they were plotted (calculated)?
Author Response
Reviewer 3:
Comments and Suggestions for Authors
The paper "Spatial transcriptomics reveals novel mechanisms involved in perineural invasion in pancreatic ductal adenocarcinomas" is a well-conducted study that employs a range of state-of-the-art instrumental techniques. As a result, all findings are well supported by the experimental data.
1) I have one recommendation regarding the organization of the manuscript: Given the numerous abbreviations used in the paper, I suggest creating a separate "Abbreviations" section. This would help readers understand and navigate the paper more easily.
Response: We acknowledge the high number of abbreviations. We had defined abbreviations when they first appear in the text. In the revised version, we have followed the journal instructions that Acronyms/Abbreviations/Initialisms should be defined the first time they appear in each of three sections: the abstract; the main text; the first figure or table. When defined for the first time, the acronym/abbreviation/initialism has been be added in parentheses after the written-out form.
And few minor comments:
2) Line 174: Leica (?).
Response: Spelling has been corrected.
3) Lines 177, 182: p.H., probably pH (?)
Response: Spelling has been corrected.
4) Line 213: of interest (!)
Response: We have changed the text. Revised text Line 216 now reads: Additionally, single-cell detection was performed based on DAPI-stained nuclei segmentation followed by application of thresholding algorithm to classify detected cells for MGLL or Nestin positivity in PNI and non-PNI regions.
- 1, b. No arrows and arrowheads are seen.
Response: We have edited the figure and figure legend to clarify. In the original arrows were below Figure1b to show morphological staining, now Figure 1f.
- S6, S7. Please, describe, what are the straight lines, how they were plotted (calculated)?
Response: We have added further details in the legend of Figures S6 and S7 to clarify what is plotted.
Added information: Scatterplots of normalised gene expression of significant differentially expressed Ligand/Receptor in PNI and non-PNI regions were identified using BulkSignalR (adjusted p value <0.20, Supplementary Table 9 or 10) with superimposed linear model line. The trend line best represents the relationship of expression between receptor and ligand in PNI or non-PNI regions.